# Sedentary Patterns and Sit-to-Stand Transitions in Open Learning Spaces and Conventional Classrooms among Primary School Students

**DOI:** 10.3390/ijerph19138185

**Published:** 2022-07-04

**Authors:** Jani Hartikainen, Eero A. Haapala, Arja Sääkslahti, Anna-Maija Poikkeus, Taija Finni

**Affiliations:** 1Faculty of Sport and Health Sciences, University of Jyväskylä, 40014 Jyväskylä, Finland; eero.a.haapala@jyu.fi (E.A.H.); arja.saakslahti@jyu.fi (A.S.); taija.m.juutinen@jyu.fi (T.F.); 2Institute of Biomedicine, University of Eastern Finland, 70211 Kuopio, Finland; 3Department of Teacher Education, University of Jyväskylä, 40014 Jyväskylä, Finland; anna-maija.poikkeus@jyu.fi

**Keywords:** sedentary behavior, physical activity, school, open learning spaces, sit-to-stand transitions

## Abstract

Educational reforms worldwide have resulted in schools increasingly incorporating open and flexible classroom designs that may provide possibilities to reduce sedentary behavior among students during lessons. Cross-sectional associations of classroom type on accelerometry assessed sedentary bout durations and sit-to-stand transitions were investigated in 191 third and fifth grade students recruited from one school with open learning spaces and two schools with conventional classrooms. A three-way ANOVA for classroom type, gender and grade level indicated that students in open learning spaces had more 1-to-4-min sedentary bouts (mean difference 1.8 bouts/h, *p* < 0.001), fewer >10-min sedentary bouts (median 0.20 vs. 0.48 bouts/h, *p* = 0.004) and more sit-to-stand transitions (mean difference 0.9 STS/h, *p* = 0.009) than students in conventional learning spaces. Comparisons between schools by grade, which were conducted with a one-way ANCOVA adjusted for gender, indicated that most of the significant differences occurred between schools with different classroom types. There were only small and mostly statistically nonsignificant differences between the two schools with conventional classrooms. In conclusion, open learning spaces may improve children’s sedentary profiles towards shorter sedentary bout durations and facilitate also postural transitions during lessons, which may translate into beneficial health impacts over a longer period.

## 1. Introduction

International physical activity guidelines recommend an average of 60 min/day of moderate-to-vigorous intensity aerobic physical activity, regular muscle-strengthening activity and a reduction in sedentary behavior, such as prolonged sitting [1,2]. Higher levels of physical activity, defined as bodily movement produced by skeletal muscles that results in increased energy expenditure, have been associated with better cardiometabolic, vascular, bone and mental health in children [3,4]. Decreasing sedentary behavior, defined as an energy expenditure of ≤1.5 metabolic equivalents of task while being awake in a sitting or reclining posture [5], and shorter duration of sedentary bouts may confer health benefits in children and youth [6,7]. Experimental studies have suggested that both short bouts of physical activity and frequent interruptions in sitting have beneficial effects on cardiometabolic biomarkers, which may reduce the risk for type 2 diabetes and metabolic syndrome in children [6,8].

Despite the evidence for the benefits of promotion of physical activity and reducing sedentary behavior, a substantial proportion of children globally grow increasingly sedentary and do not attain the recommended levels of daily physical activity [9,10]. In school settings, European primary school children aged 10–12 years have been reported to spend 65 to 70% of their school time sedentary and approximately 5% in moderate-to-vigorous physical activity, with boys accumulating less sedentary time and more moderate-to-vigorous physical activity than girls [11].

Schools are seen as feasible sites for interventions that aim to reduce sedentary time and increase overall physical activity because children spend a large proportion of their waking hours at school [12]. Lessons taking place in general education classrooms have received increasing attention as possible settings to influence children’s daily physical activity in addition to physical education and recess [13]. During lessons it is possible to reduce and break up children’s prolonged sedentary behavior by multiple different classroom-based strategies, such as physically active lessons and active breaks with or without curriculum content [14]. However, some studies have suggested that classroom-based physical activity interventions yield mostly small or no effects on physical activity and sedentary behavior [15]. Therefore, alternative approaches to reduce the sedentary behavior of students are warranted.

The affordances for physical activity provided by the indoor built environments of schools are not yet well understood, although some studies have suggested that radical changes in the architecture and furniture of a classroom may increase physical activity and reduce sedentary behavior [16]. Active school design has been shown to have some beneficial effects on sedentary behavior and light intensity physical activity but not on moderate-to-vigorous physical activity [17]. Furthermore, the elements of flexible learning spaces, including adjustable furniture with multiple uses combined with student-centered pedagogies, have been shown to facilitate positive changes in adolescents’ sedentary profiles during class time; for instance, the number of breaks in sitting (i.e., postural transitions from sitting to another posture) has been reported to be greater in flexible learning spaces compared to traditional classrooms [18].

At the same time that general education classrooms have received increasing attention as possible settings to influence children’s daily physical activity [13], schools have increasingly incorporated non-partitioned, open, flexible designs and instructional approaches that foster student autonomy, self-regulated learning, collaboration and digital competences [19]. In Finland, conventional self-contained classrooms have increasingly been replaced by more flexible, multipurpose, informal and transformable open learning spaces, in particular, after the most recent curriculum reform of Finnish basic education was issued in 2016 [20,21]. Open learning spaces may enhance opportunities for classroom-based physical activity among students to the extent that the goals set for open learning spaces bear resemblance to activity permissive classrooms [22] and flexible learning spaces [18] with multipurpose and adaptable spaces for movement.

The physical, social and cultural landscapes of a school influence teaching practices [23] and working in open learning spaces usually also implies a redistribution of teachers’ roles and responsibilities towards teams sharing space and resources [19]. The affordances and pedagogical methods enabled by open and flexible learning spaces encourage teachers to utilize more interactive teaching and collaborative learning with an emphasis on professional co-planning [24,25]. However, adaptation to novel spaces has been shown to be demanding, and teachers have faced new challenges. These include balancing between facilitating autonomous student learning and managing of shared spaces and resources in their pedagogical practice, difficulties in changing one’s institutional routines, creating coherent pedagogy for an open learning space, potential clashes between the teaching team and insufficient teachers’ skills for manipulating the environment [19,20,24,26,27,28].

Despite the expected benefits of open and flexible classrooms, we have previously observed that students’ engagement in open learning spaces may involve a surprisingly high proportion of sedentary time but more breaks from sedentary time during lessons compared to conventional classrooms [29,30]. Students have been observed to be sedentary 55–68% of classroom time, which equals 33 to 41 min of sedentary time per 60 min spent in classroom [29,30]. An increased number of breaks from sedentary time despite the higher sedentary time may indicate that sedentary time is accumulated in shorter bouts. Therefore, in the present study involving a comparison between open and conventional learning spaces, accelerometry-assessed sedentary patterns were investigated with postural transitions from sitting to standing. To examine the potential differences between schools rather than between different classroom types, we investigated the differences among three schools: one with open learning spaces and two with conventional classrooms.

## 2. Materials and Methods

### 2.1. Study Design and Participants

This cross-sectional study was conducted using data collected in years 2018–2019 in the *Children’s Physical Activity Spectrum: Daily Variations in Physical Activity and Sedentary Patterns Related to School Indoor Physical Environment* (CHIPASE) study. The University of Jyväskylä Ethics Committee approved the research protocol. Third and fifth grade students and their parents (or legal guardians) were provided with a plain language study description and consent form. Both the students’ and parents’ (or legal guardians’) consents were obtained from a total of 206 participants.

The CHIPASE data collection has been previously described in [30,31]. Fifteen classrooms of third and fifth grade students from three public schools from two different provinces in Finland participated in this study. First permissions were obtained from school principals and teachers, after which students were recruited on a voluntary basis. The school with open learning spaces participated in our previous study [29]. The two schools representing conventional school designs were chosen so that they had similar number of students for both of the grade levels recruited for this study. Third graders were chosen as the youngest grade level recruited for this study because this was the youngest age level in open learning spaces (grade 1–2 students attended conventional classrooms). Fifth grade students were chosen as the other age grade level because fifth graders participate in the national physical functional capacity monitoring and feedback system for Finnish students (MOVE!, https://www.oph.fi/en/move (accessed on 20 June 2022)). MOVE! data were collected as part of a larger research project investigating the associations between open learning space and functional capacity in children. 

One of the schools contained separate open learning spaces for each grade level from third to sixth, where the students attending third and fifth grades (70–80 students in each grade) had most of their lessons. A collective teacher team of three teachers was responsible for teaching the student group of each grade. Each grades’ open learning space contained a large space with mobile furniture that afforded multiple options for classroom activity, as well as a quiet work room (Figure 1). The students did not have a designated desk for them in the open learning spaces. In other two schools, the students attended most of their lessons in conventional classrooms with designated desks (Figure 2). One teacher was responsible for teaching a classroom of 20–25 students in the conventional schools.

Each class was assessed once during one school week. Accelerometers were distributed to be used by the students continuously during the measurement week on Monday. The students kept a diary of accelerometer wear time and absences from school during the week of measurement with assistance from their parents or legal guardians. Both the diaries and accelerometers were collected back from the participants at end of the measurement week on Friday. The classroom teachers provided a curriculum of activities for the week and the contents of the instruction followed the curriculum of the grades in question and was not in any way altered by the researchers.

### 2.2. Accelerometry Outcomes

Classroom-based sedentary patterns were assessed by waist-worn accelerometers, while postural transitions from sitting to standing (sit-to-stand transitions) were assessed with an accelerometer attached on the mid-anterior thigh. The waist-worn accelerometers are positioned near the center of the mass of the human body and, therefore, are thought to best reflect the movement of the whole body [32]. The thigh-worn accelerometers can be used to assess posture and, therefore, also to separate sitting or lying down from standing and physical activity [32,33]. Triaxial accelerometers (RM42, UKK Terveyspalvelut Oy, Tampere, Finland, Range ±16 g, sample rate: 100 Hz, A/D conversion: 13-bit) were used.

Accelerometer data reduction methods have been previously described in [29,30,31]. The teacher-reported weekly schedule was used to determine time spent inside in the classroom during general education, which was included in the analysis. Physical education and recess were excluded from the analysis. The students’ diaries were used to exclude possible absences from school, for example, due to illness. The accelerometer data were visually inspected for each lesson for each participant separately to ensure that the accelerometers were worn as reported by the participants.

For assessment of sedentary patterns, the mean amplitude deviation (MAD) method was used, as it utilizes universal g values instead of arbitrary counts, and it has been shown to be an accurate method across different accelerometer brands [34,35]. For waist-worn accelerometers, the MAD was calculated from the resultant acceleration in non-overlapping 1 s epochs on the supercomputer of CSC, the Finnish IT Center for Science. The MAD values were averaged over 15-s intervals to capture short bursts of physical activity [36] with MATLAB R2018a (The MathWorks Inc., Natick, MA, USA). The cut-off for sedentary behavior was determined as 16.7 mg, which has been previously used in assessing school-aged children in Finland [35,37]. To quantify sedentary patterns, the number of sedentary bouts were calculated for the following categories: 1-to-4 min, 5-to-9 min, 10-to-19 min, 20-to-29 min and ≥30 min [38]. Sedentary bouts of less than one minute were excluded, and a sedentary bout was considered to end with any interruption in sedentary time [39]. Sit-to-stand transitions were assessed using a thigh-worn accelerometer, attached on the thigh, with the sit-to-stand transition algorithm [40] using MATLAB (R2019a, The MathWorks Inc., Natick, MA, USA). To account for any differences in wear time of the accelerometers during classroom time, outcome variables were calculated in proportion to 60 min of classroom time [18].

### 2.3. Statistical Analyses

Descriptive statistics reported as means and standard deviations were calculated using Microsoft Excel (Microsoft Corporation, Redmond, WA, USA). Further statistical analyses were carried out using R 4.0.5 (R Studio Team, Boston, MA, USA). The normality of the data distribution was assessed using normal Q–Q plots, histograms and the Shapiro–Wilk test (*p* < 0.05). Variables violating the normality assumption were treated with a log(x + 1) transformation to meet the requirements of the normality distribution. Homogeneity of the variances was assessed using a residuals vs. fitted values plot and Levene’s test (*p* < 0.05) for all outcome variables. For variables violating the homogeneity of variance, the heteroskedasticity-consistent HC3 version of Huber–White’s robust standard errors were used.

A three-way factorial ANOVA (2 × 2 × 2) with Type III Sum of Squares, implemented with R-package car [41], was used to examine associations of the type of classroom (open vs. conventional), grade (third vs. fifth grade) and gender (boys vs. girls) on outcome variables. To examine differences between schools rather than between classroom types, comparisons were made for both grade levels separately with a one-way ANCOVA, with gender set as the covariate. The statistical significance was set at *p* ≤ 0.05 with 95% confidence intervals. Tukey’s honest significance test was utilized for multiple comparisons.

## 3. Results

### 3.1. Descriptive Statistics

A total of 204 students participated in the assessments, and waist-worn accelerometry was obtained from 197 students. After excluding participants with missing thigh-worn accelerometer data, the final sample size was reduced to 191 students. Table 1 displays the means and standard deviations of the accelerometer outcomes across the three participating schools and two grade levels.

### 3.2. Associations of Gender, Grade Level and Classroom Type on Sedentary Behavior

A three-way factorial ANOVA was used to examine the three- and two-way interaction and main effects of gender (girls vs. boys), grade level (fifth grade vs. third grade) and classroom type (open vs. conventional) on different sedentary bout duration categories and sit-to-stand transitions. Due to the small observed number of bouts >10-min, the sedentary bout categories of 10-to-19-min and 20-to-29 min were combined for the three-way ANOVA analysis. Sedentary bouts lasting over 30-min were not observed. Table 2 shows results of the three-way ANOVA test of between-subjects effects of grade, gender and classroom type on sedentary behavior variables.

Statistically significant three-way interactions between gender, grade and classroom type on the sedentary behavior variables were not observed (Table 2). A significant two-way interaction was observed between gender and classroom type on the 1-to-4-min sedentary bouts (Table 2). However, the post hoc test indicated that the differences between boys and girls were not significant in either open learning spaces or conventional classrooms. Both girls (mean difference 1.2 bouts/h, *p* = 0.003) and boys (mean difference 2.4 times/h, *p* < 0.001) had more 1-to-4-min sedentary bouts in the open learning spaces compared to the conventional classroom, when the means were adjusted for the grade level (Table 3). The main effect of classroom type on 1-to-4-min sedentary bouts was significant (Table 2), as the students in open classrooms had more 1-to-4-min bouts (mean difference 1.8 bouts/h, *p* < 0.001) than in the conventional classrooms when the means were adjusted for grade level and gender (Table 3).

For the 5-to-9-min sedentary bouts, neither two-way interactions nor main effects were observed (Table 2). For >10-min bouts, assumptions of the normality of data and the homogeneity of variance were not met, and the number of >10-min bouts per hour was first log(x + 1)-transformed. After log(x + 1)-transformation, Levene’s test still indicated a violation of the homogeneity of variances. Therefore, a robust ANOVA was conducted using the HC3-version of Huber–White’s robust standard errors, which indicated that there was a significant two-way interaction between gender and grade (Table 2).

The post hoc test indicated that fifth grade girls had more >10-min bouts than third grade girls (median; interquartile range: 0.60; 0.50 vs. 0.31; 0.41 bouts/h, *p* = 0.004) when adjusted for classroom type. The main effect for classroom type was significant, and the students in open learning spaces had fewer >10-min sedentary bouts (median; interquartile range: 0.20; 0.24 vs. 0.48; 0.55 bouts/h, *p* < 0.001), when adjusted for grade level and gender (Table 3).

For sit-to-stand transitions, a robust three-way ANOVA was conducted as the assumption for the homogeneity of variance was not met. A significant main effect for classroom type was observed, as the students in conventional classrooms had fewer sit-to-stand transitions (0.9 STS/h, *p* = 0.009) compared to the students in open learning spaces when the means were adjusted for grade level and gender.

### 3.3. Grade-Matched Differences between Schools

A one-way ANCOVA was used to investigate differences in the sedentary behavior variables between schools controlled for gender. There were significant differences between schools (F(2,108) = 14.816, *p* < 0.001) in the 1-to-4-min sedentary bouts in the third grade students. The third grade students in school A had more 1-to-4-min bouts than their counterparts in schools B (mean difference 1.5 bouts/h, *p* < 0.001) and C (mean difference 1.7 bouts/h, *p* < 0.001) (Table 4). Significant differences were observed also for the fifth grade students between schools (F(2,75) = 14.801, *p* < 0.001). The fifth grade students in school A had more 1-to-4-min sedentary bouts than the students in schools B (mean difference 1.6 bouts/h, *p* = 0.011) and C (mean difference 2.5 bouts/h, *p* < 0.001) (Table 4).

For the 5-to-9-min bouts, the differences between schools were not significant either for the third or fifth grade students. The estimated marginal means adjusted for gender indicated similar numbers of 5-to-9-min bouts in all three schools in both grade levels (Table 4). For log(x + 1)-transformed >10-min sedentary bouts, there were significant differences between schools in the third grade students (F(2,108) = 8.634, *p* < 0.001). The third grade students in school A had fewer >10-min sedentary bouts compared to schools B (median; interquartile range: 0.20; 0.20 vs. 0.36; 0.65 bouts/h, *p* = 0.011) and C (median; interquartile range: 0.42; 0.45 bouts/h, *p* = 0.012) (Table 4). In the fifth grade students, covariate gender was significantly associated with >10-min sedentary bouts (F(1,75) = 5.598, *p* = 0.021). However, there was an overlap between 95% confidence intervals of log(x + 1)-transformed estimated marginal means of with >10-min sedentary bouts between girls (95%CI [0.35, 0.50]) and boys (95%CI [0.23; 0.38]). There were also statistically significant differences between schools (F(2,75) = 4.773, *p* = 0.11). The fifth grade in school A had fewer >10-min sedentary bouts than the students in school B (median; interquartile range: 0.27; 0.33 vs. 0.67;0.22 bouts/h, *p* = 0.013), while the differences between schools A-C and B-C were not statistically significant (Table 4).

In third grade, statistically significant differences were observed for the students’ sit-to-stand transitions between schools (F(2,108) = 12.198, *p* < 0.001). The third grade students in school A had more sit-to-stand transitions than the students in school C (mean difference 2.6 transitions/h, *p* < 0.001), and there was also a statistically significant difference between schools B and C (mean difference 1.9 transitions/h, *p* = 0.011) (Table 4). In fifth grade, statically significant differences between schools were not observed for the students’ sit-to-stand transitions (Table 4).

## 4. Discussion

The present study investigated third and fifth grade students’ accelerometry-assessed sedentary patterns and postural transitions from sitting to standing between open and conventional learning spaces and between three schools. The results indicated that the students in open learning spaces had more 1-to-4-min sedentary bouts, fewer >10-min sedentary bouts and more sit-to-stand transitions than the students in conventional learning spaces. There were no differences in 5-to-9-min bouts between the open learning spaces and conventional classrooms. In line with previous research [17,18], the current results indicate that sedentary time is accumulated in open and flexible learning spaces in shorter bouts with more frequent breaks in sedentary time and more postural transitions. Therefore, open learning spaces may provide potential benefits by breaking up the prolonged sedentary time of school-aged children and youth [6,7,8]. Some differences also occurred between the two schools with conventional learning spaces in the sit-to-stand transitions among third grade students, but the differences between the conventional schools were modest and statistically not significant. Although school level policies and individual teacher’s pedagogical practices may influence the accumulation and breaking up of sedentary time [42], the present study suggests that classroom type seems to exert a greater influence than school on classroom-based sedentary behavior.

Gender and grade level had an interaction effect on >10-min sedentary bouts as fifth grade girls had more >10-min sedentary bouts than third grade girls. These finding are consistent with previous findings, indicating that older students, especially girls, tend to be more sedentary than younger students [11,43,44,45,46]. These findings suggest that interventions targeting classroom-based sedentary behavior need to focus on reducing sedentary behavior among older students, especially among girls. Furthermore, when examining classroom physical activity interventions, the gender and grade level or age of the participants should be considered.

Strengths of this present study include the use of accelerometry-derived measures of classroom-based sedentary behavior in authentic settings where teaching methods were not experimentally altered. This approach enabled estimation of the associations of classroom type on classroom-based sedentary behavior in real life conditions. Furthermore, our statistical approach allowed analysis of the potential associations of participants’ gender and grade level on classroom-based sedentary behavior. Potential differences between schools, in addition to classroom type, were also investigated.

Limitations of this study include its cross-sectional nature, which excludes confirmation of any causal relationships between the assessed variables. Furthermore, our sample size of 15 classes and an unbalanced design including one school with open learning space and two schools with conventional classroom, reduces the statistical power and possibilities for clustering students within classes and schools with sophisticated approaches, such as hierarchical linear modeling [47]. We did not control for the possible influences of weight, body fat content or anthropometry on classroom-based sedentary behavior because such procedure is quite rare in epidemiological settings. However, we acknowledge that children who are overweight have been observed to spend significantly less time in moderate-to-vigorous intensity activities than children with normal weight [11]. For instance, one study found that while children of normal weight in the intervention group were more active than children of normal weight in the control group, similar differences were not observed among overweight and obese children [48]. Therefore, future studies are needed to examine whether associations between the type of classroom learning environment and classroom-based sedentary behavior are different in populations of normal and overweight children.

School-level physical activity policies were not assessed, but all three schools participated in the national action program, Finnish Schools on the Move, which aims to establish a physically active culture in Finnish comprehensive schools. Approximately 90% of Finnish elementary schools and 95% of pupils are involved in the program [49]. Schools and municipalities that participate in the program implement their own plans to enhance physical activity during physical education, recess and academic lessons [49,50], and, thus, there may be some differences in the activities performed during the school week that were not controlled for in this study. For example, if students participate in vigorous physical activity during physical education or recess, they may be less physically active during the classroom lessons. It is also possible that teachers feel that breaking up students’ sedentary time is less necessary if students have already been physically active during the PE lesson or recess. Information is currently limited on the relation between sedentary and physical activity in different contexts, in particular, on how the extent of activity in different lessons, and during recess and lunch time, influence each other [51].

As the physical aspects of learning spaces do not influence sedentary behavior alone, but exert their influence together with factors related to the school culture and pedagogical solutions [23], future studies should investigate potential school-level policies and potential teachers’ intrapersonal factors, such as their perceptions of the value of physical activity [42], which were not included in this study. Furthermore, this study did not involve assessments of students’ experiences regarding open learning spaces compared to conventional classrooms. However, a recent study indicated that students studying in learning spaces with flexible furniture have reported greater satisfaction with the learning environment than students in classrooms with traditional furniture, as the former provides more opportunities for student autonomy [52]. Students’ attending open and flexible learning spaces have been observed to engage more in collaborative learning activities and to incorporate mobility into their own learning activities, while developing agency by choosing how and where they will work [25]. Open and flexible classroom designs can influence social relationships by facilitating spontaneous interactions among students and teachers [25]. There is some evidence that academic results in English, Mathematics and Humanities may benefit from the utilization of flexible learning spaces in Australian children and adolescents [53]. Associations between open learning spaces and academic results have not been studied in the Finnish educational setting. Therefore, future studies should seek to investigate the potential effects of open learning spaces on the academic results of Finnish primary school-aged children.

Finally, the accelerometer data reduction methods and the accelerometers themselves used in this study are somewhat different than those in prior studies [18,38], and, therefore, the results of the different studies are not directly comparable. Currently, there is no clear consensus about the most valid operational definitions of accelerometer-based measures among researchers [39]. The MAD method used for assessing accelerometer data in this study has documented validity and reliability across different accelerometer brands [34,35]. The sit-to-stand transition algorithm has been shown to be reliable in free-living environments in community-dwelling older adults [40], but it has not been yet validated for children.

## 5. Conclusions

Students in open learning spaces were found to have more 1-to-4-min sedentary bouts, fewer >10-min sedentary bouts and more sit-to-stand transitions, while there were no differences in the 5-to-9-min sedentary bouts between open learning spaces and conventional classrooms. Shorter sedentary bouts and more postural transitions may induce health benefits in school-age children in the long term. Studies with longitudinal multi-level approaches are warranted.

## Figures and Tables

**Figure 1 ijerph-19-08185-f001:**
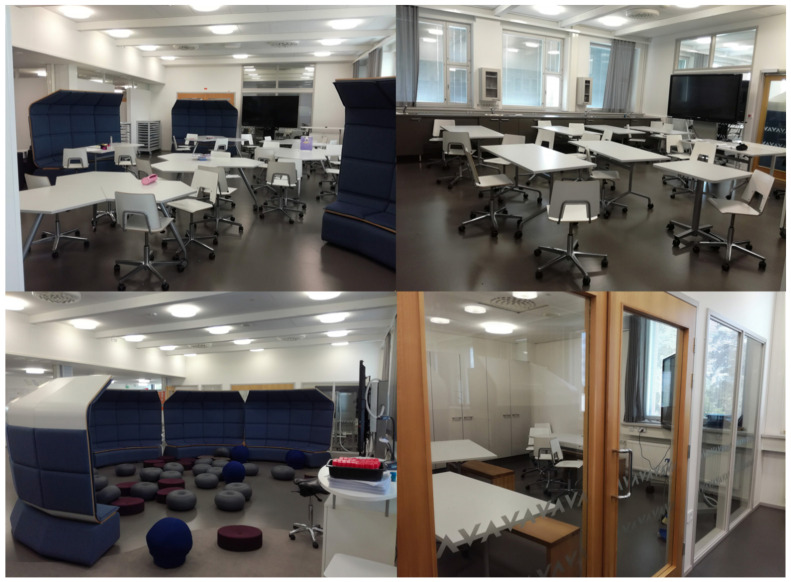
The open learning space shows several areas for work as well as a quiet work room allowing for division of the class of about 70–80 students into smaller groups with mobile and dynamic furniture.

**Figure 2 ijerph-19-08185-f002:**
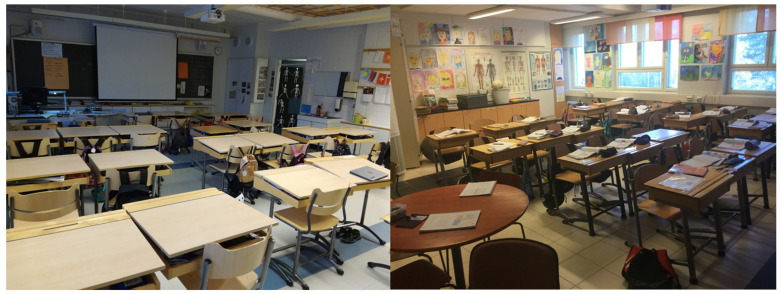
Pictures of the conventional classrooms represent typical, smaller, self-contained rooms for around 20 students with a designated desk for each student.

**Table 1 ijerph-19-08185-t001:** Results of the sedentary behavior assessments by school and grade level.

School	A		B		C	
Classroom type	Open		Conventional		Conventional	
Grade level	3rd	5th	3rd	5th	3rd	5th
Number of participants	38	21	52	33	22	25
Girls (%)	42.1	52.4	59.6	51.5	50.0	48.0
1–4 min Sedentary bouts (bouts/h)	6.80 ± 1.27	6.78 ± 1.99	5.32 ± 1.57	5.13 ± 1.64	5.10 ± 1.41	4.27 ± 1.09
5–9 min Sedentary bouts (bouts/h)	1.51 ± 0.60	1.59 ± 0.68	1.38 ± 0.49	1.58 ± 0.49	1.42 ± 0.51	1.49 ± 0.45
>10 min Sedentary bouts (bouts/h)	0.20 ± 0.15	0.31 ± 0.31	0.39 ± 0.33	0.60 ± 0.43	0.42 ± 0.28	0.52 ± 0.26
10–19 min Sedentary bouts (bouts/h)	0.19 ± 0.15	0.25 ± 0.27	0.35 ± 0.29	0.52 ± 0.38	0.38 ± 0.28	0.41 ± 0.26
20–29 min Sedentary bouts (bouts/h)	0.00 ± 0.02	0.07 ± 0.09	0.04 ± 0.08	0.08 ± 0.12	0.04 ± 0.07	0.11 ± 0.15
30+ min Sedentary bouts (bouts/h)	0	0	0	0	0	0
Sit-to-stand transitions (bouts/h)	6.54 ± 1.84	5.41 ± 2.52	5.77 ± 2.19	5.32 ± 1.66	3.93 ± 1.57	4.65 ± 1.47

Means and standard deviations. Girls (%) describes percentage of girls in subsamples.

**Table 2 ijerph-19-08185-t002:** Three-way ANOVA test of between-subjects effects of grade, gender and classroom type on sedentary behavior variables.

	F(7,183)						
Sedentary Behavior Variable	Gender	Grade	Classroom	Gender x Grade	Gender x Classroom	Grade x Classroom	Gender x Grade x Classroom
1–4 min Sedentary bouts	2.244	0.723	54.380 ***	2.643	5.940 *	1.062	0.160
5–9 min Sedentary bouts	0.171	1.442	0.957	0.069	0.525	0.232	0.009
>10 min Sedentary bouts ^a,b^	3.566	9.000 **	22.686 ***	4.612 *	0.032	0.227	0.216
Sit-to-Stand Transitions ^b^	0.144	3.289	5.174 *	0.567	0.526	1.572	0.549

* *p* < 0.05, ** *p* < 0.01, *** *p* < 0.001, ^a^ log(x + 1) transformation was utilized. ^b^ Three-way ANOVA was conducted using the heteroskedasticity-consistent HC3 version of Huber–White’s robust standard errors.

**Table 3 ijerph-19-08185-t003:** Estimated marginal means of post hoc analyses after three-way ANOVA.

1-to-4-min Sedentary Bouts (Bouts/h) Adjusted for Grade Level				
** *Gender* **	** *Classroom Type* **	** *Estimated Marginal Mean* **	** *Lower CI95%* **	** *Upper CI95%* **
Girls	Open	6.3	5.7	6.9
Boys	Open	7.3	6.7	7.8
Girls	Conventional	5.1	4.8	5.5
Boys	Conventional	4.9	4.5	5.2
**1-to-4-min sedentary bouts (bouts/h) adjusted for grade level and gender**				
** *Classroom type* **		** *Estimated marginal mean* **	** *Lower CI95%* **	** *Upper CI95%* **
Open		6.8	6.4	7.2
Conventional		5.0	4.7	5.3
**>10-min bouts (log(bouts/h + 1)) ^a^ adjusted for classroom type**				
** *Gender* **	** *Grade* **	** *Estimated marginal mean* **	** *Lower CI95%* **	** *Upper CI95%* **
Girls	5th	0.40	0.33	0.48
Boys	5th	0.27	0.20	0.35
Girls	3rd	0.24	0.18	0.30
Boys	3rd	0.25	0.19	0.30
**>10-min bouts (log(bouts/h + 1)) ^a,b^ adjusted for grade level and gender**				
** *Classroom type* **		** *Estimated marginal mean* **	** *Lower CI95%* **	** *Upper CI95%* **
Open		0.21	0.16	0.27
Conventional		0.37	0.33	0.40
**Sit-to-Stand-transitions (transitions/h) ^b^ adjusted for grade level and gender**				
** *Classroom type* **		** *Estimated marginal mean* **	** *Lower CI95%* **	** *Upper CI95%* **
Open		6.0	5.5	6.6
Conventional		5.1	4.8	5.5

^a^ log(x + 1) transformation was utilized. ^b^ Three-way ANOVA was conducted using the heteroskedasticity-consistent HC3 version of Huber–White’s robust standard errors.

**Table 4 ijerph-19-08185-t004:** Grade-matched between school-estimated marginal means of sedentary behavior variables controlled for gender.

School—Classroom Type	Significant Difference between Schools	Estimated Marginal Mean	Lower CI95%	Upper CI95%
**1-to-4-min sedentary bouts (Bouts/(h)**				
** *3rd grade* **				
A—Open	A-B ***, A-C ***	6.8	6.4	7.3
B—Conventional		5.3	4.9	5.7
C—Conventional		5.1	4.5	5.7
** *5th grade* **				
A—Open	A-B *, A-C ***	6.8	6.1	7.5
B—Conventional		5.1	4.6	5.7
C—Conventional		4.3	3.6	4.9
**5-to-9-min sedentary bouts (bouts/h)**				
** *3rd grade* **				
A—Open		1.5	1.3	1.7
B—Conventional		1.4	1.2	1.5
C—Conventional		1.4	1.2	1.7
** *5th Grade ^b^* **				
A—Open		1.6	1.4	1.8
B—Conventional		1.6	1.4	1.8
C—Conventional		1.5	1.3	1.7
**>10-min bouts (log(bouts/h+1)) ^a^**				
** *3rd Grade ^b^* **				
A—Open	A-B *, A-C *	0.17	0.11	0.24
B—Conventional		0.30	0.25	0.36
C—Conventional		0.33	0.25	0.41
** *5th Grade ^b^* **				
A—Open	A-B *	0.25	0.16	0.35
B—Conventional		0.43	0.36	0.51
C—Conventional		0.41	0.32	0.50
**Sit-to-Stand Transitions (transitions/h)**				
** *3rd Grade* **				
A—Open	A-C ***	6.5	5.9	7.2
B—Conventional	B-C *	5.8	5.2	6.3
C—Conventional		3.9	3.1	4.8
** *5th Grade ^b^* **				
A—Open		5.4	4.6	6.2
B—Conventional		5.3	4.7	6.0
C—Conventional		4.7	3.9	5.4

^a^ log(x + 1) transformation was utilized. ^b^ One-way ANCOVA was using the heteroskedasticity-consistent HC3 version of Huber–White’s robust standard errors. * *p* < 0.05, *** *p* < 0.001.

## Data Availability

The data presented in this study are available upon reasonable request from the corresponding author.

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
