# Peer review of "Sedentary Patterns and Sit-to-Stand Transitions in Open Learning Spaces and Conventional Classrooms among Primary School Students"

_ijerph, 2022, doi:10.3390/ijerph19138185_

Round 1

Reviewer 1 Report

I really liked your article, and I found it useful even for those teaching at the university level.  I especially appreciate the attention to gender discussion of the study's strengths and weaknesses.

I was interested, though, in whether students liked or learned more in open-learning vs. traditional classrooms.  While these questions were not within the purview of your study, you might suggest them for further research.  

I caught some mechanical/grammatical errors as follows:

page 2, line 69--has reported

page 2, line 87--has been shown

page 2, line 94--students' engagement

page 3, line 102--differences among three schools

page 3, line 105--Study design

page 5, line 139--content of instruction

page 10, line 311--older students and

Author Response

We thank you for kind feedback regarding our manuscript. We have now included some discussion on students’ experiences and academic results regarding open and flexible learning spaces. We thank you for noticing the grammatical errors and we have now corrected them. Please see highlighted changes in the revised manuscript.

Reviewer 2 Report

Engaging in physical activity results not only in increased energy expenditure, but has also been associated with better cardiometabolic, vascular, and mental health in children and adults. However, despite the promotion of physical activity, a high percentage of children do not meet the physical activity recommendations. The following comments are sent to the authors for their consideration:

1. How were the schools selected? Were these public or private?

2. Why choose only 3rd and 5th graders? Why didn't they include younger or older children?

3. Why were physical education and recess time excluded? Could these data be a confounding factor? That is, the children of the 2 schools with conventional classes could carry out more activity during these times since they have been sitting for more time.

4. Table 1. Authors are suggested to indicate in the table if there is a statistically significant difference between the schools. Were differences found between the two conventional schools?

5. Table 2. The gender variable, are they men or women? Is the grade variable 3rd or 5th? It is not identified in the results to which category it belongs.

6. The authors in the analysis make a stratification of the minutes, it would be appropriate to generate a variable where the sum of the total minutes of sedentary lifestyle is.

7. Table 4. The table does not show the p-values ​​of difference between the groups.

8. Physical education classes and recess within the school are important moments that should be used to activate children, that is, schools should, as far as possible, adapt and promote active spaces for children. However, what can the authors mention in the discussion regarding the responsibility of parents to engage in physical activity outside or inside the home (not at school)?

Author Response

Comment: Engaging in physical activity results not only in increased energy expenditure, but has also been associated with better cardiometabolic, vascular, and mental health in children and adults. However, despite the promotion of physical activity, a high percentage of children do not meet the physical activity recommendations. The following comments are sent to the authors for their consideration:

Response: We would like to thank you for your insightful comments, which have helped us to improve our manuscript.  Please find point by point responses for your comments below.

Comment 1: How were the schools selected? Were these public or private?

Response:  The school with open learning spaces had participated in our previous study while two schools representing conventional school designs were chosen so that they had similar number of students for both grade levels recruited for this study. All schools were public. Please see revised manuscript lines 114-120:  

“Fifteen classrooms of 3rd and 5th grade students from three public schools from two different provinces in Finland participated in this study. First permissions were obtained from school principals and teachers, after which students were recruited on a voluntary basis. The school with open learning spaces had participated in our previous study [29]. Two schools representing conventional school designs were chosen so that they had similar number of students for both grade levels recruited for this study. “

Comment: 2. Why choose only 3rd and 5th graders? Why didn't they include younger or older children?

Response: Third graders were chosen the as youngest grade-level recruited for this study because this was the youngest age level in open learning spaces (grade 1-2 students attended conventional classrooms). Fifth grade students were chosen as the other age grade level because fifth graders participate in the national physical functional capacity monitoring and feedback system for Finnish students. MOVE! data were collected as part of a larger research project investigating associations of open learning space with functional capacity in children. Please see revised manuscript lines 120-127.

“Third graders were chosen the as youngest grade-level recruited for this study because this was the youngest age level in open learning spaces (grade 1-2 students attended conventional classrooms). Fifth grade students were chosen as the other age grade level because fifth graders participate in the national physical functional capacity monitoring and feedback system for Finnish students (MOVE!, https://www.oph.fi/en/move).  MOVE! data were collected as part of a larger research project investigating associations of open learning space with functional capacity in children.”

Comment: 3. Why were physical education and recess time excluded? Could these data be a confounding factor? That is, the children of the 2 schools with conventional classes could carry out more activity during these times since they have been sitting for more time.

Response: We thank you for pointing this out. Physical education lessons and recess time were excluded as aim of our study was to investigate classroom-based sedentary behavior in open learning spaces and conventional classrooms during general education lessons. However, the reviewer correctly notes that physical activity carried out during physical education or recess may influence classroom-based sedentary time. We have now included a comment on this relevant point in discussion. Please see lines 373-386:

School-level physical activity policies were not assessed, but all three schools participated in the national action program, Finnish Schools on the Move, aiming to establish a physically active culture in Finnish comprehensive schools. Approximately 90% of Finnish elementary schools and 95% of pupils are involved in the program [49]. Schools and municipalities participating in the program implement their own plans to enhance physical activity during physical education, recess, and academic lessons [49,50], and, thus, there may be some differences in the activities performed during the school week which were not controlled in this study. For example, if students participate in vigorous physical activity during physical education or recess, they may be less physically active during the classroom lessons. It is also possible that teachers feel that breaking up students’ sedentary time is less necessary if students have already been physically active during the PE lesson or recess. Information is currently limited on the relation between sedentary and physical activity in different contexts, in particular on how extent of activity in different lessons, and during recess and lunch time influence each other [51]. “

Comment: 4. Table 1. Authors are suggested to indicate in the table if there is a statistically significant difference between the schools. Were differences found between the two conventional schools?

Table 1 describes means and standard deviations of accelerometer outcomes across the three participating schools and two grade levels. Because of unequal gender distribution between groups, differences between schools across two different grade levels were adjusted for gender and the results are described later in section 3.3 and Table 4. Findings concerning differences between the two conventional schools are presented in section 3.3.

Comment: 5. Table 2. The gender variable, are they men or women? Is the grade variable 3rd or 5th? It is not identified in the results to which category it belongs.

Response: Table 2 presents interaction and main effects of gender (girls vs. boys), grade (5th vs. 3rd), and classroom type (open vs. conventional). Thus, table 2 provides information on whether statistically significant overall differences were found across the groups. Potential differences regarding pairwise comparisons were analyzed with post hoc analyses, which are explained in text and shown in Table 3.

Comment: 6. The authors in the analysis make a stratification of the minutes, it would be appropriate to generate a variable where the sum of the total minutes of sedentary lifestyle is.

Response: Findings on amount of classroom-based sedentary time have been reported in a recently published article and these finding have now been included in introduction. Please see lines 93-100:

“Despite the expected benefits of open and flexible classrooms, we have previously observed that students’ engagement in open learning spaces may involve surprisingly high proportion of sedentary time but more breaks from sedentary time during lessons compared to conventional classrooms [29,30]. Students have been observed to be sedentary 55-68% of classroom time, which equals 33 to 41 minutes of sedentary time per 60 minutes spent in classroom [29,30]. Increased number of breaks from sedentary time despite higher sedentary time may indicate that sedentary time is accumulated in shorter bouts”

Comment: 7. Table 4. The table does not show the p-values ​​of difference between the groups.

Response: Thank you for pointing this out. Table 4 was updated to include p-values concerning comparisons between schools.

Comment: 8. Physical education classes and recess within the school are important moments that should be used to activate children, that is, schools should, as far as possible, adapt and promote active spaces for children. However, what can the authors mention in the discussion regarding the responsibility of parents to engage in physical activity outside or inside the home (not at school)?

Response: The reviewer points out correctly that the role and responsibility of parents is important regarding their children’s physical activity. However, this theme was somewhat outside of the scope of our study which focused on different types of school settings and therefore after careful deliberation we decided not to include parents’ role in the discussion. 

Reviewer 3 Report

Thank you for the opportunity of reviewing this paper.

In my opinion it is a well written, interesting manuscript showing important results about an emerging and promising theme such as school active breaks.

I have only two suggestions for the Authors. First, they should report the literature references for the methodology used for accelerometry (including why those body sites were chosen). Second, they should include a consideration about possible influence of weight, body mass index, or anthropometry on physical activity levels reached by pupils. See some Italian experience about (i.e. doi: 10.3390/ijerph17186599).

Author Response

We thank you for your kind feedback. We have now updated references with respect to methodology used for accelerometry and included description of why the specific body sites were chosen. Please see highlighted changes in section 2.2. Accelerometry Outcomes 

We thank you for suggesting a consideration about the possible influence of weight, body mass index, or anthropometry on students’ physical activity levels. We have now included a comment on this in the discussion. Please see lines 362-372:

“We did not control for possible influence of weight, body fat content, or anthropometry on classroom-based sedentary behavior because such procedure is quite rare in epidemiological settings. However, we acknowledge that children with overweight have been observed to spend significantly less time in moderate to vigorous intensity activities than children with normal weight [11]. For instance, one study found that while normal weight children in the intervention group were more active than normal weight children in the control group, similar differences were not observed among children with overweight and obesity [48]. Therefore, future studies are needed to examine whether associations between type of classroom learning environment and classroom-based sedentary behavior are different in populations of normal and over-weight children.”

Round 2

Reviewer 2 Report

The authors have responded and made the necessary changes to the manuscript.